# Improved EEG-Based Emotion Classification via Stockwell Entropy and CSP Integration

**DOI:** 10.3390/e27050457

**Published:** 2025-04-24

**Authors:** Yuan Lu, Jingying Chen

**Affiliations:** National Engineering Research Center for E-Learning, Central China Normal University, Wuhan 430079, China; ly21@jmu.edu.cn

**Keywords:** EEG, emotion recognition, Stockwell transform, Stockwell entropy, common spatial pattern

## Abstract

Traditional entropy-based learning methods primarily extract the relevant entropy measures directly from EEG signals using sliding time windows. This study applies differential entropy to a time-frequency domain that is decomposed by Stockwell transform, proposing a novel EEG emotion recognition method combining Stockwell entropy and a common spatial pattern (CSP). The results demonstrate that Stockwell entropy effectively captures the entropy features of high-frequency signals, and CSP-transformed Stockwell entropy features show superior discriminative capability for different emotional states. The experimental results indicate that the proposed method achieves excellent classification performance in the Gamma band (30–46 Hz) for emotion recognition. The combined approach yields high classification accuracy for binary tasks (“positive vs. neutral”, “negative vs. neutral”, and “positive vs. negative”) and maintains satisfactory performance in the three-class task (“positive vs. negative vs. neutral”).

## 1. Introduction

Emotion recognition refers to the detection and analysis of human emotional states through technological means [1]. The electroencephalogram (EEG) technique measures brain activity by recording the electrical activity of cortical neurons [2]. Due to its non-invasive nature and real-time capabilities, EEG technology plays a crucial role in brain function research, particularly in studies that require the prolonged continuous monitoring of brain activity [3]. EEG signals are commonly represented in the form of waveforms, where distinct waveforms, such as alpha, beta, theta, and delta waves, correspond to different states of brain activity [4]. By analyzing key features of EEG signals, including frequency, power spectral density, and phase, researchers are able to investigate the functional state of the brain and discern various emotional states, such as happiness, sadness, and anxiety [5]. As a result, the potential of EEGs as a tool for emotion recognition is considered highly promising [6].

However, due to the asymmetric and non-stationary nature of electroencephalogram (EEG) signals [7], emotion recognition based on EEGs remains a complex scientific challenge [8]. Recently, learning approaches based on entropy measures have been demonstrated to be one of the most effective technical pathways for emotion-related EEG recognition [9]. Entropy, as a measure of information uncertainty, possesses a strong capability to extract clinically meaningful regularity information from EEG signals [10]. Entropy measures can be utilized to quantify the irregularity, randomness, and complexity of physiological signals [9]. Moreover, there is substantial and compelling evidence indicating that entropy-based metrics are highly effective for analyzing EEG signals [9,11]. For instance, Beatriz García-Martínez and Arturo Martínez-Rodrigo et al. were the first to introduce the application of three entropy-based metrics, namely, sample entropy, quadratic sample entropy, and distribution entropy, to differentiate between calm and negative-stress emotional states [12]. Ruo-Nan Duan and Jia-Yi Zhu et al. proposed differential entropy (DE) features and compared them with traditional energy spectrum (ES) features, demonstrating that DE features and their associated combinations offer superior performance for emotion recognition [11]. Yun Lu and Mingjiang Wang et al. introduced a dynamic entropy-based pattern-learning framework for recognizing inter-individual emotions from EEG signals, aiming to address the poor generalization capability of existing emotion recognition methods, which is caused by individual variability [9]. Wei-Long Zheng and Bao-Liang Lu utilized differential entropy features extracted from multi-channel EEG data to train a deep belief network for identifying positive, neutral, and negative emotions. They also investigated the weights of the trained deep belief network to explore key frequency bands and channels [13]. Research by Wu et al. demonstrates that Stockwell entropy, combined with the Hilbert transform, effectively detects events of interest (EoIs) compared to the standard Hilbert transform, achieving accurate identification of both EoIs and non-EoIs [14]. The experimental results indicated that neural features related to different emotions do, indeed, exist and exhibit commonalities across different experiments and individuals [13].

In this study, we propose combining Stockwell differential entropy with the CSP algorithm for the classification and recognition of emotional EEG signals. Specifically, differential entropy is first applied to the time-frequency domain decomposed by the Stockwell transform, followed by the use of the CSP algorithm to extract feature vectors corresponding to different emotional states. Additionally, this study further investigates how signal frequency and amplitude influence Stockwell entropy and its classification performance.

## 2. Related Work

The successful implementation of any EEG-based emotion recognition system relies on the accurate identification of features representing emotional states, which necessitates efficient feature extraction algorithms [15]. The extracted features must exhibit high discriminability to achieve superior recognition rates. The Stockwell transform is a time-frequency analysis tool that combines the advantages of short-time Fourier transform (STFT) and wavelet transform analyses, providing a detailed view of frequency variations over time [16,17]. CSP is another feature extraction technique that is widely used in biomedical applications. CSP employs a linear transformation method to project multi-channel EEG data into a low-dimensional space, thereby enhancing its discriminative capability for classifying EEG data across different categories [18,19]. K. Venu and P. Natesan proposed an approach called the HC+SMA-SSA scheme, which extracts features using improved Stockwell transform (ST) and CSP for motor imagery task classification, demonstrating superior performance in key metrics [20]. S. Sethi and R. Upadhyay et al. proposed a feature extraction method that integrates the Stockwell transform technique with CSP for designing motor imagery-based brain-computer interfaces, significantly improving the discriminability of motor imagery tasks [21]. Mausovi et al. combined wavelet transform (WT) with the CSP algorithm, achieving high classification accuracy in asynchronous offline brain–computer interface applications [22]. M.I. Chacon-Murguia and E. Rivas-Posada evaluated five feature extraction methods, including Stockwell, CSP + CWT, and CSP + ST, for classifying two types of motor imagery signals. Their proposed methods demonstrated superior performance compared to conventional approaches [23].

The main contributions of this paper are organized as follows: Section 3 introduces the EEG dataset used in this study, along with the Stockwell transform, Stockwell entropy, the CSP algorithm, and the experimental procedure. Section 4 provides a detailed comparison of the emotion recognition performance of the combined Stockwell entropy and CSP approach across different frequency bands and emotional states, presenting both data analysis and experimental results. Section 5 offers an in-depth analysis of the experimental outcomes and the underlying reasons for this observed classification performance. The final section summarizes the key findings and conclusions of this study.

## 3. Dataset and Methods

### 3.1. Dataset

The SEED dataset is a publicly available dataset provided by Shanghai Jiao Tong University, which was primarily designed for research in affective computing [13]. This dataset comprises electroencephalogram (EEG) signals collected from 15 participants (7 males and 8 females), recorded using a 62-channel EEG system, arranged according to the international 10–20 standard, and covering major regions of the brain. During the experiments, the participants watched 15 carefully selected film clips (encompassing positive, neutral, and negative emotions), each lasting approximately 4 min, to induce coherent emotional responses. Consequently, each participant’s data file contains 15 segments of preprocessed EEG data (channels × time-series data). The experiments were conducted three times for each participant, with intervals of approximately 1 week or longer between sessions [13].

### 3.2. Preprocessing

This study utilizes the “Preprocessed_EEG” brain electrical data files provided by the SEED dataset, as these EEG data files have already been downsampled and preprocessed. The data has been downsampled to 200 Hz and filtered using a 0–75 Hz bandpass filter [13]. On this basis, this study further applied a 0.1–46 Hz bandpass filter.

### 3.3. Methods

#### 3.3.1. Stockwell Transform

Stockwell transform, also known as the S-transform, combines the advantages of the Fourier transform and the wavelet transform, providing a means for achieving the multi-resolution analysis of signals. It is particularly well-suited for the analysis of non-stationary signals [24,25].

Stockwell transform can be mathematically defined as follows:$$S(\tau ,f)=\int_{-\infty }^{+\infty }x(t)w(\tau -t)e^{-j2\pi ft}dt$$

In the given expression, *x(t)* denotes the original signal, and the window function *w(t)* is typically a Gaussian window. Here, *τ* represents the time position, and *f* represents the frequency. This integral expression provides the complex value of the signal at a given time and frequency, while the magnitude represents the energy level at that specific time and frequency [26,27].

The formula demonstrates that the Stockwell transform technique exhibits multi-resolution analysis capabilities [27]. Through the time-frequency decomposition of non-stationary signals, the technqiue generates a time-frequency representation of the signal [28]. The spectrum characteristics of the signal vary over time, thereby enabling the visualization of frequency component variations at different time points [29]. Unlike the short-time Fourier transform (STFT), the S-transform employs a frequency-dependent Gaussian window function, comprising a narrower window for high frequencies and a wider window for low frequencies [27,30]. Consequently, at high frequencies, the S-transform achieves higher time resolution through a narrower time window, while at low frequencies, it maintains better frequency resolution using a wider time window [31]. This adaptive window function enables the S-transform to more accurately capture transient features in non-stationary signals, making it a valuable tool for analyzing such signals in numerous fields including EEG and ECG [32,33,34].

#### 3.3.2. Stockwell Transform Entropy

Shannon entropy serves as a measure of information uncertainty. Time-frequency signals processed using the Stockwell transform technique yield a new feature vector by computing the differential entropy in the time domain for each frequency band through a sliding time window approach. This feature vector is termed “Stockwell entropy” [14].

By calculating Stockwell entropy, we can quantify the complexity and randomness of EEG signals across different frequency bands over certain time periods. The computational process of Stockwell entropy using a sliding window approach is depicted in Figure 1.

Under the assumption that the time series ***X*** follows a Gaussian distribution $N(\mu ,\sigma^{2})$, the formula for differential entropy shows that, for a fixed-length EEG sequence, the differential entropy in a specific frequency band equals the logarithm of the energy spectrum [35]:(1)$$f(X)=-\int_{-\infty }^{\infty }p(x)\cdot \log(p(x))dx=\frac{1}{2}\log(2\pi e\sigma^{2})$$
where: $p(x)=\frac{1}{\sqrt{2\pi \sigma^{2}}}e^{-\frac{{\left(x-\mu \right)}^{2}}{2\sigma^{2}}}$

Suppose that in a certain epoch, the EEG signal of one channel, after employing Stockwell transform, has a time-series of length *N* in the time-frequency signal, denoted as $S=[s_{1},s_{2},\ldots ,s_{N}]$, and a sliding window with a width of *k*, where k≤N. Then, the data processing process of the sliding window on the time-series in the time-frequency signal is as follows.

For each position *i*, where 1≤i≤N−k+1, the sub-sequence covered by the sliding window is $S_{i}=[s_{i},s_{i+1},\ldots ,s_{i+k-1}]$.

Step 1. Calculate the average value of all elements within the window *S_i_*.

Step 2. Calculate the standard deviation of the sub-sequence covered by the sliding window *S_i_*.

Step 3. Apply Formula (1) to each sub-sequence covered by the sliding window, to obtain a new sequence $T=[t_{1},t_{2},\ldots ,t_{N-k+1}]$. The new formula can be expressed as follows:$$T=f([s_{i},s_{i+1},\ldots ,s_{i+k-1}])=\frac{1}{2}\log(2\pi e\sigma_{i}^{2})$$

The obtained sequence $T=[t_{1},t_{2},\ldots ,t_{N-k+1}]$ is Stockwell entropy.

#### 3.3.3. Common Spatial Pattern

The CSP is a feature extraction algorithm that highlights spatial distribution patterns in the EEG signals associated with specific tasks [36,37]. The core principle of the CSP algorithm involves finding a set of spatial filters that maximize the variance between two types of trials [38]. When these spatial filters are applied to the original EEG signals, the signals are projected into a new feature space, generating features that are optimized for classification. In this transformed space, the samples from different categories become more distinguishable [39,40]. Additionally, CSP reduces the dimensionality of the original EEG data by extracting discriminative features while preserving classification-relevant information. Finally, the extracted feature vectors are fed into machine learning classifiers, such as support vector machines (SVM) or random forests, for training and testing to classify the different task states. In this study, Stockwell entropy is processed using the CSP algorithm to generate eight distinct feature vectors for subsequent analysis [40].

#### 3.3.4. Feature Extraction Process Combining Stockwell Entropy and CSP

To obtain certain features of EEG activity based on Stockwell entropy and CSP, in this study, the EEG time periods for one experiment for each subject (including 15 EEG segments) were first decomposed into several epochs. In this study, the length of each epoch was set to 3 s, and each epoch served as a sample, corresponding to an emotional state code (1—positive, 0—neutral, and −1—negative). Subsequently, time-frequency decomposition based on Stockwell transform was performed on each channel of every epoch [41]. For this paper, the ‘epochs.compute_tfr’ function from the MNE library was used with the following parameters: ‘method = “stockwell”’, ‘freqs = (0.1, 46.0)’, and ‘width = 1’ [42]. Then, the coefficients of the Stockwell transform were grouped, and the time-frequency domain was divided into six different frequency sub-bands, namely, the full frequency band (0.1, 46) Hz, Delta (0.1, 4) Hz, Theta (4, 8) Hz, Alpha (8, 12) Hz, Beta (12, 30) Hz, and Gamma (30, 46) Hz frequency bands. Next, the differential entropy value was calculated for each “frequency sub-band-time” unit on each channel of every epoch. The result of this step was to obtain a new “channel × frequency-sub-band × Stockwell entropy” matrix. Subsequently, the CSP algorithm was executed separately on each frequency sub-band to obtain the corresponding emotional feature vectors of each frequency sub-band on each channel [41]. At the same time, the effect of dimensionality reduction was also achieved. Finally, the generated feature vectors were input into classifiers such as SVM or random forest for the classification of emotional states. Figure 2 shows the flowchart of the Stockwell entropy-CSP feature extraction method.

### 3.4. Classification

In the first step of this study, stratified k-fold cross-validation was adopted to evaluate the effectiveness of the “Stockwell entropy–CSP combination model”; at the same time, this could avoid data leakage in the second-step classification comparison. This study used the most common stratified 5-fold cross-validation technique, dividing the training dataset into 5 subsets. In each iteration, a different subset was used as the test set, and the remaining part was used as the training set. Firstly, the “positive, neutral, and negative” emotions of each subject were combined. Three groups of experiments, namely, “Positive vs. Neutral”, “Positive vs. Negative”, and “Neutral vs. Negative”, respectively, were designed to verify the model’s performance. Meanwhile, the influence of different frequency bands on the classification of emotional states was examined. Then, the frequency band with the best evaluation effect was tested under different window states and for different emotional classifications. The final result of the experiment was an average recognition rate of 15 subjects in the three groups of experiments.

In the second step, two classifiers—SVM and random forest (RF)—were employed to compare the model’s classification performance on the test set data. SVM performs exceptionally well in high-dimensional spaces and can effectively address nonlinear classification problems using kernel functions. In contrast, random forest, as an ensemble learning method, demonstrates strong robustness against noise and overfitting [42]. By comparing the recognition rates of these two classifiers, the stability and reliability of the classification results were comprehensively validated. Four experimental setups were designed: Positive vs. Neutral vs. Negative, Positive vs. Neutral, Positive vs. Negative, and Neutral vs. Negative. In each setup, the data were randomly divided into training and test sets, with 80% being allocated to the training set and the remaining 20% to the test set. The final results were reported as the average recognition rates across the four experimental groups for all 15 subjects.

## 4. Results

### 4.1. Impact of Different Frequency Bands on Emotional State Binary Classification

As described in Section 3.4, this experiment employed stratified five-fold cross-validation. Three experimental setups—Positive vs. Neutral, Positive vs. Negative, and Neutral vs. Negative—were examined to assess the impact of different frequency bands on emotional state classification. The impact of different frequency bands is evaluated for each experimental setup. The classifier employed here is an RBF-kernel SVM, with its parameters set to kernel = ‘rbf’, C = 20, and Gamma = ‘scale’.

Table 1 summarizes the recognition accuracy and standard deviation obtained using stratified five-fold cross-validation for three binary classification tasks: Positive vs. Neutral, Negative vs. Neutral, and Positive vs. Negative. The results are reported across different window widths (W_5, W_10, W_20) and frequency bands (full frequency band, Delta, Theta, Alpha, Beta, and Gamma). Abbreviations are defined as follows: Pos = positive, Neu = neutral, and Neg = negative.

Figure 3 illustrates the recognition accuracy of pairwise classifications among three emotion categories (Positive vs. Neutral, Negative vs. Neutral, Positive vs. Negative) across different frequency bands: the full frequency band, Delta, Theta, Alpha, Beta, and Gamma. The *x*-axis denotes the frequency bands, while the *y*-axis indicates the recognition accuracy. Specifically, Figure 3a corresponds to a sliding window width of 5 during Stockwell entropy calculation, Figure 3b corresponds to a sliding window width of 10, and Figure 3c corresponds to a sliding window width of 20.

Table 1 and the three images in Figure 3 indicate that the Gamma frequency band achieved the highest accuracy across all three emotional combinations, with the full frequency band following closely. The classification accuracies of the Delta and Theta frequency bands were generally low, indicating significant differences in classification performance across the different frequency bands and tasks. Selecting an appropriate frequency band is crucial for enhancing the accuracy of EEG-based emotion classification.

### 4.2. Influence of Gamma Frequency Band on the Classification of Four Emotional Combinations

Since the Gamma frequency band exhibited the highest accuracy, the classification performance of this band was further evaluated for four emotional combinations: three-class classification (Positive, Negative, and Neutral), Positive vs. Neutral, Negative vs. Neutral, and Positive vs. Negative. This evaluation was conducted under different sliding window widths (W_5, W_10, and W_20). The classifier employed was an RBF-kernel SVM with the parameters set as follows: kernel = ‘rbf’, C = 20, and Gamma = ‘scale’, as shown in Figure 4. The *x*-axis denotes the window width, while the *y*-axis indicates the recognition accuracy. W_5, W_10, and W_20 correspond to sliding window widths of 5, 10, and 20, respectively, during Stockwell entropy calculation.

Figure 4 shows that, for the Gamma frequency band, increasing the window width from W_5 to W_20 had minimal impact on classification accuracy, thereby demonstrating the relatively stable performance of the Stockwell entropy-CSP algorithm in EEG-based emotion recognition. The accuracy of the Gamma frequency band in two-class classification tasks (Positive vs. Neutral, Negative vs. Neutral, and Positive vs. Negative) was generally high, with the lowest accuracy reaching 92.8%. Moreover, the accuracies of Positive vs. Neutral and Positive vs. Negative both remained above 96%. When the window width was 20, the classification accuracy of Positive vs. Neutral reached its highest value of 96.8%. Generally, the CSP algorithm is more suitable for two-class classification scenarios. However, as shown in Figure 4, the CSP algorithm was also applied to evaluate the three-class classification case (Positive vs. Negative vs. Neutral). Although the three-class classification accuracy was lower than that for the two-class case, the lowest level of accuracy still reached 88.7%.

As shown above, as the window width increased from 5 to 20, the model’s classification accuracy tended to improve. However, increasing the window width had a limited impact on classification accuracy. Most classification tasks demonstrated consistent performance across all window widths. For two-class tasks (Positive vs. Neutral, Negative vs. Neutral, and Positive vs. Negative), high accuracy was achieved at all window widths. The three-class classification task (Positive vs. Negative vs. Neutral) achieved lower accuracy than the two-class tasks but remained above 88% across all window widths.

### 4.3. Influence of Different Classification Methods on Emotional State Recognition

In the first two sections, the model’s performance was evaluated using stratified five-fold cross-validation. In this section, the classification performance of two classifiers—RBF-SVM (kernel = ‘rbf’, C = 20, Gamma = ‘scale’) and random forest (n_estimators = 256)—on the test set data was compared. In the experiment, the emotional states of each subject (positive, neutral, and negative) were used to design four experimental setups: Positive vs. Neutral vs. Negative, Positive vs. Neutral, Positive vs. Negative, and Neutral vs. Negative. In each setup, the data were randomly divided into training and test sets, with 80% of the data allocated to the training set and the remaining 20% to the test set. The final results were reported as the average recognition rates across the four experimental groups for all 15 subjects.

The experimental results demonstrate that the varying window widths have minimal impact on classification accuracy. Figure 5 compares the classification accuracy across different frequency bands for binary tasks (Positive vs. Neutral, Negative vs. Neutral, and Positive vs. Negative) and the ternary task (Positive vs. Negative vs. Neutral). The analysis was performed using a sliding window width of 20. The *x*-axis denotes the frequency bands (full frequency band, Delta, Theta, Alpha, Beta, and Gamma), while the *y*-axis indicates the classification accuracy. Figure 5 reveals that the overall performance of SVM and RF was similar across all frequency bands. RF outperformed SVM in the Delta, Theta, and Alpha bands, whereas SVM achieved slightly higher accuracy in the Gamma band. Notably, the Gamma band yielded the highest accuracy under both classifiers. Binary classification tasks generally achieved greater accuracy than ternary tasks. Nevertheless, both classifiers exceeded 88% accuracy in the Gamma band for ternary classification tasks.

## 5. Discussion

### 5.1. Experimental Conclusions

This study’s innovative nature is demonstrated by applying differential entropy to the time-frequency domain decomposed by the Stockwell transform and by proposing an EEG-based emotion extraction method that integrates Stockwell entropy with the CSP algorithm.

The experiments demonstrate the following:The classification accuracy was highest in the Gamma frequency band.Increasing the sliding window width from W_5 to W_20 had a minimal impact on classification accuracy, demonstrating that the Stockwell entropy–CSP algorithm exhibits relatively stable performance in EEG-based emotion recognition.The accuracy of binary classification tasks—namely, Positive vs. Neutral, Negative vs. Neutral, and Positive vs. Negative—was generally high. Among these tasks, Positive vs. Neutral and Positive vs. Negative achieved the highest recognition rates.Although the CSP algorithm is more suitable for binary classification tasks, it also demonstrated strong performance in the three-class task (Positive vs. Negative vs. Neutral).

In prior neurophysiological research, the overall body of evidence also supported an association between Gamma oscillations and emotional states [43]. Yang et al. (2020) observed increased Gamma connection density between the prefrontal, temporal, parietal, and occipital regions during emotional processing [44]. Luther et al. (2023) reported increased Gamma power over posterior areas for unpleasant compared to pleasant and neutral pictures [45]. These findings suggest that Gamma oscillations play a role in various aspects of emotional processing, including the perception of emotional stimuli, the cognitive regulation of emotions, and the interaction between emotions and other cognitive processes [46].

### 5.2. Analysis of the Reasons Behind the Experimental Results

#### 5.2.1. Influence of Signal Frequency and Amplitude on Stockwell Entropy

This study primarily focuses on the analysis of swept-frequency signals. To better simulate the behavior of Stockwell entropy for real EEG signals, the frequency range of the swept-frequency signal was set to 0–46 Hz, with a sampling rate of 200 Hz. Figure 6 illustrates the influence of different signal states on Stockwell entropy. Figure 6a demonstrates the effect of increasing frequency with constant amplitude on Stockwell entropy. As the frequency of the swept-frequency signal increases, the entropy values of high-frequency signals gradually stabilize, whereas those of low-frequency signals exhibit significant fluctuations. Figure 6b depicts the scenario where both signal frequency and amplitude increase. The results indicated that the entropy values of high-frequency signals increased approximately linearly and remained relatively concentrated, while those of the low-frequency signals fluctuated significantly. Figure 6c represents the scenario where signal frequency increases while amplitude decreases. It can be seen that the entropy values of high-frequency signals decreased approximately linearly and remained relatively concentrated, whereas those of low-frequency signals continued to fluctuate significantly. Collectively, Figure 6a–c demonstrates that as the frequency increases, the entropy values across different window widths converge toward stability.

As shown in Figure 6d,e, we used the controlled variable method to analyze the effects of amplitude and frequency changes on Stockwell entropy. Figure 6d illustrates the scenario where the signal amplitude increased while the frequency remained constant, with the signal frequency fixed at 5 Hz. The results indicated that when the signal frequency was held constant and the amplitude increased, the entropy value exhibited oscillatory growth, accompanied by significant fluctuations. Figure 6e depicts the scenario where the signal amplitude remained unchanged while the frequency varied. It is evident that low-frequency signals exhibit substantial fluctuations in entropy values, whereas high-frequency signals demonstrate relatively stable behavior.

Table 2 compares the standard deviations of Stockwell entropy for the signal sin(2πft) across different frequencies and window widths (W = 5, 10, 20). The table lists only the standard deviations at the boundary frequencies of different frequency bands. The results indicate that as the frequency increased, the standard deviations generally decreased. When the frequency reached the Gamma band, the standard deviations reported in Table 2 typically dropped below 1, thereby showing that Stockwell entropy values are relatively stable and suitable for effective emotional classification. Figure 6e compares the results for frequencies of 13 Hz and 30 Hz.

From Figure 6 and Table 2, the following conclusions can be drawn:The entropy values of high-frequency signals are relatively stable, while those of low-frequency signals fluctuate significantly.Both an increase and a decrease in amplitude can cause changes in the entropy values of high-frequency signals, and these changes are approximately linear. Therefore, the entropy values of high-frequency signals respond well to amplitude changes and can be used to detect such changes.As the frequency increases, the values under different window conditions tend to stabilize, indicating that the selection of window width has little impact on the Stockwell entropy values of high-frequency signals. This demonstrates that Stockwell entropy values are highly stable for classification and recognition.

These conclusions indicate that the entropy values in the high-frequency Gamma band remain relatively stable, thereby contributing to the identification of emotional states. This is why the classification accuracy in this band is relatively high.

#### 5.2.2. Influence of the CSP Algorithm on Emotional State Classification

In this study, the CSP algorithm was applied to decompose Stockwell entropy into eight components. This study uses the CSP component features of the “Positive vs. Negative” binary classification state in the Gamma frequency band of the 15th subject as an example. For clarity, Figure 7 displays the curves of three representative components. It can be observed from Figure 7 that negative emotions correspond to low values on the orange curve and high values on the blue curve, whereas positive emotions correspond to high values on the orange curve and low values on the blue curve. Thus, the CSP-transformed Stockwell entropy features exhibit high discriminability for emotional EEG signals, effectively distinguishing between different emotional states and achieving high recognition rates.

### 5.3. Deficiencies and Prospects

In this study, an emotion recognition method combining Stockwell entropy and CSP was compared with the findings of previous studies by Ruo-Nan Duan, Wei-Long Zheng, and Bao-Liang Lu. Ruo-Nan Duan and Jia-Yi Zhu investigated the classification accuracy of four features—the energy spectrum (ES), differential entropy (DE), differential asymmetry (DASM), and rational asymmetry (RASM)—for the Positive vs. Negative binary classification task. The average classification accuracies of these features were 76.56%, 84.22%, 80.96%, and 83.28%, respectively [11]. In contrast, the classification accuracies of the SVM and RF classifiers in the Gamma frequency band for the same task reached 96.1% and 95.7%, respectively. Wei-Long Zheng, Bao-Liang Lu, and their colleagues developed an EEG-based emotion recognition model using deep belief networks (DBNs). The DBNs were trained on differential entropy features extracted from multi-channel EEG data. The SEED dataset was utilized to classify three emotional states: positive, neutral, and negative, achieving a maximum accuracy of 86.65%. The average classification accuracies of DBN, SVM, logistic regression (LR), and K-nearest neighbors (KNN) were 86.08%, 83.99%, 82.70%, and 72.60%, respectively [13]. In this study, the classification accuracies of the SVM and RF classifiers for the three-class task in the Gamma frequency band reached 88.5% and 88.0%, respectively.

Although the combination of Stockwell entropy and CSP has demonstrated strong performance in classifying emotional EEG signals, this study still has certain limitations. In existing emotion recognition research, individual differences remain a significant challenge. Emotion-related signal patterns exhibit substantial inter-individual variability, potentially achieving high performance in within-subject tests but performing poorly in cross-subject scenarios. The final experimental results reported in this study represent the average recognition rates across 15 subjects; however, the emotional data used for each classification test were derived from the same individual. Therefore, future work will focus on enabling individual-independent emotion recognition and refining the proposed method accordingly. Additionally, investigating the spatial distribution of entropy features in the brain may help identify the key cortical regions associated with emotional changes, which could serve as a promising direction for future research.

## 6. Conclusions

This study proposes a method for extracting emotional EEG features by integrating Stockwell entropy with the CSP technique. Stockwell transform, a time-frequency analysis tool, provides a localized time-frequency representation of non-stationary EEG signals, while Stockwell entropy captures entropy features in the time-frequency domain. The CSP algorithm, a widely used feature extraction method, achieves dimensionality reduction and is frequently applied to spatial filtering in binary motor imagery tasks within the brain–computer interface (BCI). This study integrates these two techniques to enhance the accuracy and stability of emotional EEG recognition.

The experimental results indicate that the proposed combined method exhibits both excellent classification performance and strong stability for emotional EEG signals in the Gamma (30–46 Hz) frequency band. The method achieves high classification accuracy in binary classification tasks, including Positive vs. Neutral, Negative vs. Neutral, and Positive vs. Negative. Furthermore, it also achieves satisfactory classification results in the three-class task (Positive vs. Negative vs. Neutral). Additionally, this study investigates and analyzes the impact of varying window sizes and frequency bands on classification accuracy.

## Figures and Tables

**Figure 1 entropy-27-00457-f001:**
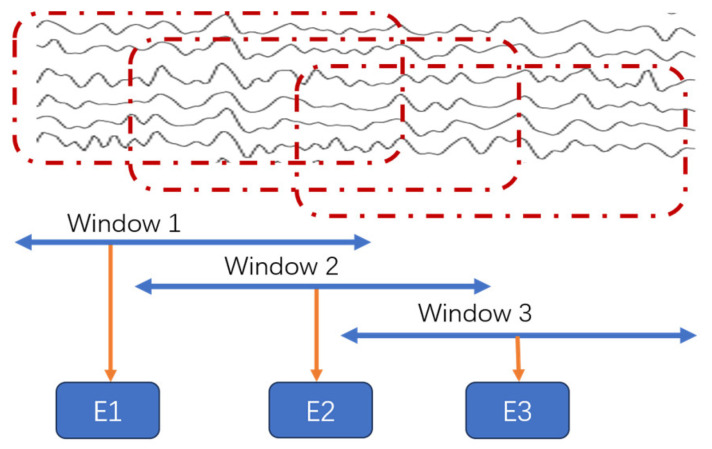
Calculation process of Stockwell entropy with sliding windows.

**Figure 2 entropy-27-00457-f002:**
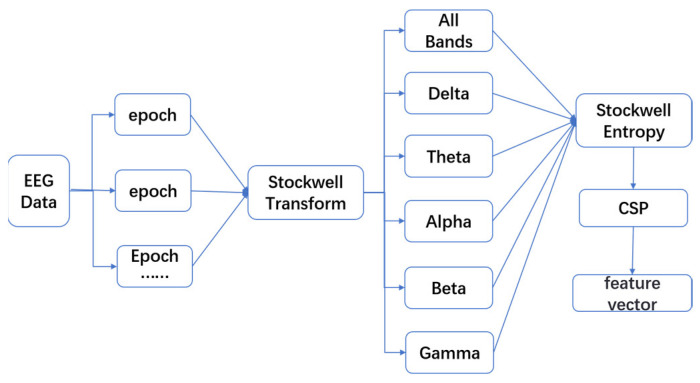
Flowchart of the Stockwell entropy-CSP feature extraction method.

**Figure 3 entropy-27-00457-f003:**
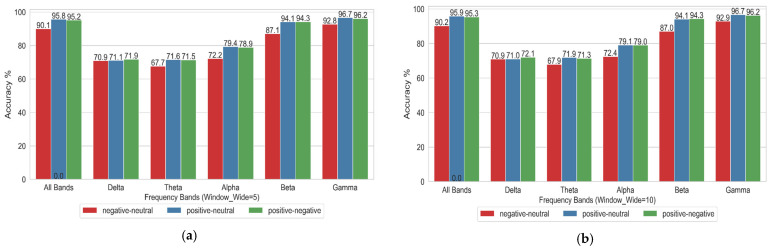
Classification of the combinations of three emotional combinations under different frequency bands: (**a**) when the width of the sliding window is 5; (**b**) when the width of the sliding window is 10; (**c**) when the width of the sliding window is 20.

**Figure 4 entropy-27-00457-f004:**
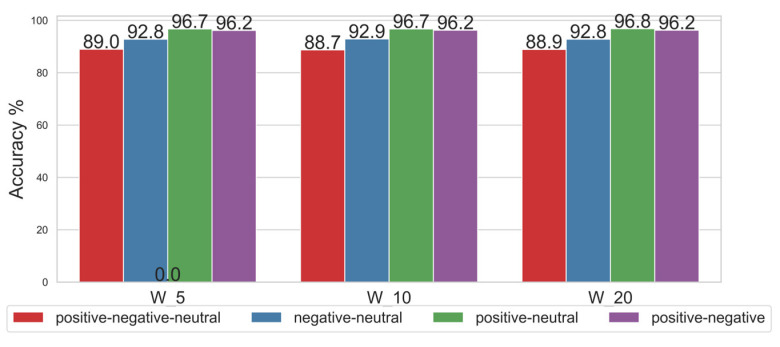
Classification of the combinations of four emotional states.

**Figure 5 entropy-27-00457-f005:**
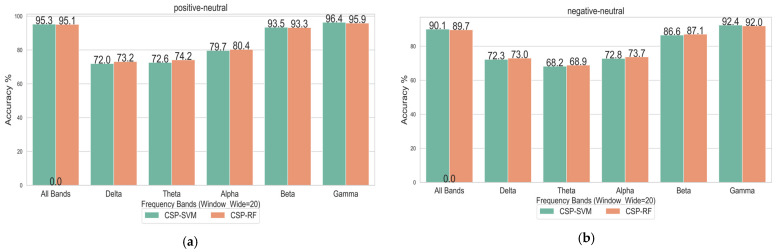
Classification of emotional states by SVM and RF in different frequency bands: (**a**) Positive vs. Neutral; (**b**) Negative vs. Neutral; (**c**) Positive vs. Negative; (**d**) Positive vs. Negative vs. Neutral.

**Figure 6 entropy-27-00457-f006:**
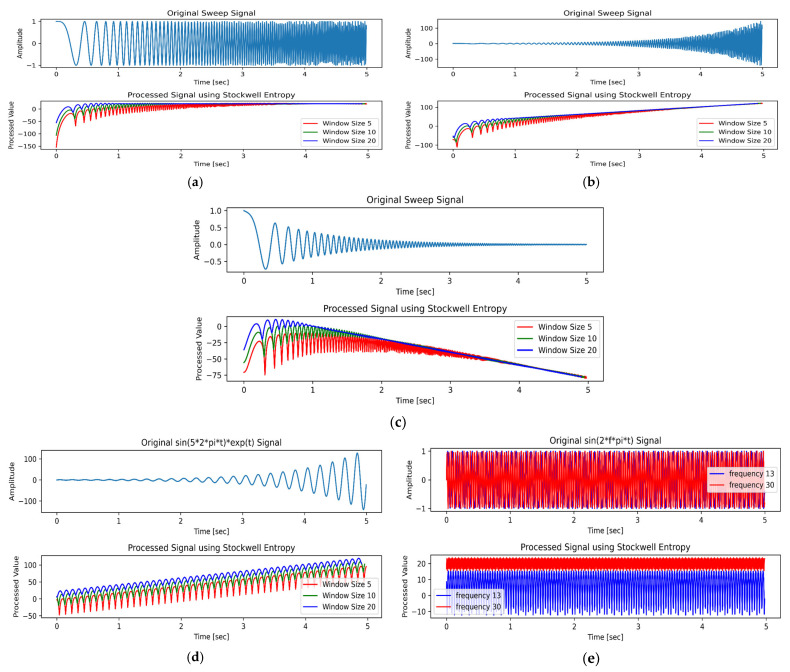
Influence of signal frequency and amplitude on Stockwell entropy: (**a**) the frequency of the swept-frequency signal increases; (**b**) both the frequency and amplitude of the swept-frequency signal increase; (**c**) the frequency of the swept-frequency signal increases, while its amplitude decreases; (**d**) the frequency of the signal is fixed, while the amplitude increases; (**e**) the amplitude of the signal is fixed, but the frequencies are different.

**Figure 7 entropy-27-00457-f007:**
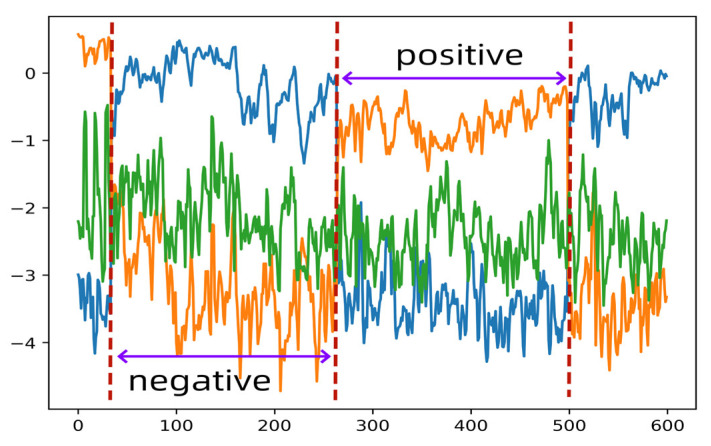
CSP components in the “Positive vs. Negative” state.

**Table 1 entropy-27-00457-t001:** Classification of three emotional combinations according to different frequency bands and different window widths.

Bands	W_5%			W_10%			W_20%		
	Neu vs. Neg	Pos vs. Neg	Pos vs. Neu	Neu vs. Neg	Pos vs. Neg	Pos vs. Neu	Neu vs. Neg	Pos vs. Neg	Pos vs. Neu
**All**	90.1 ± 1.1	95.2 ± 0.9	95.8 ± 0.8	90.2 ± 1.2	95.3 ± 0.9	95.9 ± 0.8	90.3 ± 1.2	95.5 ± 0.8	95.9 ± 0.7
**Delta**	70.9 ± 1.6	71.9 ± 1.8	71.1 ± 2	70.9 ± 1.8	72.1 ± 1.7	71 ± 1.9	70.7 ± 1.7	72.2 ± 1.6	70.9 ± 2.1
**Theta**	67.7 ± 1.9	71.5 ± 1.7	71.6 ± 1.8	67.9 ± 1.7	71.3 ± 1.6	71.9 ± 1.7	67.8 ± 2	71.1 ± 1.9	71.6 ± 1.7
**Alpha**	72.2 ± 2	78.9 ± 1.5	79.4 ± 1.6	72.4 ± 2.2	79 ± 1.6	79.1 ± 1.5	72.3 ± 2.3	79 ± 1.8	79.2 ± 1.5
**Beta**	87.1 ± 1.3	94.3 ± 1.1	94.1 ± 1	87 ± 1.3	94.3 ± 1.1	94.1 ± 1	87.1 ± 1.3	94.4 ± 0.9	94.1 ± 1
**Gamma**	92.8 ± 1.2	96.2 ± 0.9	96.7 ± 0.8	92.9 ± 1.3	96.2 ± 0.8	96.7 ± 0.7	92.8 ± 1.3	96.2 ± 0.8	96.8 ± 0.8

**Table 2 entropy-27-00457-t002:** Standard deviations of Stockwell entropy at different frequencies with different window widths.

Bands	Frequency	Window = 5 Std	Window = 10 Std	Window = 20 Std
**Delta**	1	16.62	15.43	13.66
	3	14.66	12.29	9
**Theta**	4	13.82	11.09	7.26
	7	11.93	8.17	3.04
**Alpha**	8	11.37	7.33	1.88
	12	9.39	4.4	0.76
**Beta**	13	8.94	3.75	0.59
	30	2.82	0.36	0
**Gamma**	31	2.51	0.61	0.24
	36	1.03	0.84	0.29
	41	0.23	0.22	0.21
	42	0.43	0.41	0.33
	43	0.62	0.55	0.32
	44	0.79	0.63	0.17
	45	0.93	0.64	0.04

## Data Availability

Further inquiries can be directed to the corresponding authors.

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
