# Peer review of "Improved EEG-Based Emotion Classification via Stockwell Entropy and CSP Integration"

_entropy, 2025, doi:10.3390/e27050457_

Round 1
Reviewer 1 Report (Previous Reviewer 2)
Comments and Suggestions for Authors
It is noted that almost all comments/concerns raised during the initial review have been addressed.
A few points which sought some clarification have been provided with detailed explanations.
It is recommended to relook into the citation formatting: for example, Reference 6: provide complete details, 9: why is author name without space? check all et.al format.
Comments on the Quality of English LanguageThere are many. For example:
Abstract, Line 14: "extracted by this combined method has a good classification effect" - consider changing "has" to "have" to agree with the plural "features".
Abstract, Line 15: "(30-46) Hz and have strong stability" - consider changing "have" to "has" to agree with "This method".
Abstract, Line 16: "has a high classification accuracy" - It could read as "achieves high classification accuracy".
Author Response
Thank you very much for taking the time to review this manuscript.
Comments 1: [It is recommended to relook into the citation formatting: for example, Reference 6: provide complete details, 9: why is author name without space? check all et.al format.]
Response 1: [Thank you for pointing this out. I have revised all the references.]
Comments 2:[ There are many. For example:
Abstract, Line 14: "extracted by this combined method has a good classification effect" - consider changing "has" to "have" to agree with the plural "features".
Abstract, Line 15: "(30-46) Hz and have strong stability" - consider changing "have" to "has" to agree with "This method".
Abstract, Line 16: "has a high classification accuracy" - It could read as "achieves high classification accuracy".]
Response 2:[ Thank you for pointing this out. We have thoroughly reviewed the expression throughout the entire paper and made corresponding revisions. Additionally, we have opted not to provide line-by-line annotations for the changes.]
Thank you for providing many valuable opinions.
Reviewer 2 Report (Previous Reviewer 1)
Comments and Suggestions for Authors
Thank you for your diligent effort in revising this manuscript. For the purpose of further strengthening its impact and clarity, the following refinements are suggested:
Comments 1: Preprocessing Data
Response 1: Acceptable
Comments 2: Common Spatial Patterns (CSP)
Response 2: Please provide a more detailed explanation of how the 8 CSP components were selected, adding clarification to the paragraph “3.3.3 Common Spatial Patterns” in conjunction with Figure 2.
Comments 3: Feature Extraction Process
Response 3: Acceptable
Comments 4: Classification
Response 4: Acceptable
Comments 5: Results Presentation
Response 5: Acceptable
Comments 6: Explanation of Gamma Wave Accuracy
Response 6: To enhance the credibility of the analysis regarding the link between Gamma waves and emotions, please provide supporting references. (Topic : 5.1 Experimental Conclusion)
Comments 7: Suggestions for Future Work
Response 7: Acceptable

Author Response
Thank you very much for taking the time to review this manuscript.
Comments 1: [Please provide a more detailed explanation of how the 8 CSP components were selected, adding clarification to the paragraph “3.3.3 Common Spatial Patterns” in conjunction with Figure 2.]
Response 1: [Figure 2 displays the eight optimal spatial filters computed from actual EEG data. Since Section 3.3.3 "Common Spatial Patterns (CSP)" primarily introduces the CSP algorithm, Figure 2 is not directly related to its content. This paper does not analyze these eight spatial filters, and the issue was not previously recognized. Upon reflection, we believe removing Figure 2 is more appropriate. Thank you for your suggestion.]
Comments 2: [To enhance the credibility of the analysis regarding the link between Gamma waves and emotions, please provide supporting references. (Topic : 5.1 Experimental Conclusion)]
Response 2:[Thank you for your suggestions—we've adopted them. We have incorporated references to neurophysiological research literature that supports the observational findings (Topic : 5.1 Experimental Conclusion).]
Thank you for providing many valuable opinions.
This manuscript is a resubmission of an earlier submission. The following is a list of the peer review reports and author responses from that submission.
Round 1
Reviewer 1 Report
Comments and Suggestions for Authors
Strengths
- The study demonstrates a promising approach by integrating Stockwell entropy with the Common Spatial Pattern (CSP) for emotion classification from EEG data. This combination presents a potentially valuable contribution to the field, offering a unique perspective on feature extraction.
- The manuscript is written in clear and concise language, ensuring readability and comprehension even for those whose primary language is not English.
Areas for Improvement
- Preprocessing Data
- Inconsistent Bandpass Filtering (Lines 116-117)
- Line 116-117: The paper states that the data was "filtered with a 0-75 Hz bandpass filter," which differs from reference document #13, which used "0.3-50 Hz."
- The authors should provide a detailed justification for this difference. If the deviation is intentional, the rationale and potential impact on the data should be thoroughly explained. If it was an error it must be corrected.
- Suboptimal Power Line Noise Removal (Line 118):
- Line 118: The authors claim to have used a "bandpass filter 0-46 Hz" to avoid the 50 Hz power line noise. This reasoning is flawed. The original data could have been used with the addition of a "notch filter 50 Hz."
- Common Spatial Patterns (CSP)
- Lack of Component Selection Justification (Lines 187-188):
- The authors select 8 CSP components from 62, but fail to provide a clear explanation of the selection process. This raises questions about the optimality of the chosen components and their impact on the results. In other words, the authors do not explain how the 8 CSP components were selected from the 62 components, nor whether this selection is sufficient. This creates a disconnect with Figure 8. Additionally, reference #41 uses ICA, not CSP.
- Missing Link to Figure 8:
- The link between the number of CSP components selected and the information displayed in figure 8 is not clear.
- Feature Extraction Process:
- Unexplained Epoch Length Choice (Line 195):
- The authors do not explain the rationale for choosing an epoch length of 3 seconds, especially since the source article (#13) used an epoch length of 1 second. This requires justification.
- Classification:
- Insufficient Explanation of Pairwise Classification (Lines 219-221) : The rationale for using pairwise classification across three emotion groups is not clearly explained. Readers may not understand why pairwise verification is necessary across three groups when a ternary classification of all three emotions could be used
- Results Presentation:
- Clarity in Table 1 (Line 248): Line 248, Table 1: Use abbreviations (e.g., Pos-Neg, Pos-Neu) instead of numerical codes for clarity. Also, brainwaves should be listed in order of frequency.
- Figure 4-6: Replace the bar graphs in Figures 4-6 with confusion matrices to provide a more detailed and easily interpretable representation of the classification performance.
- Discussion and Interpretation:
- Explanation of Gamma Wave Accuracy (Line 333)
- The authors report that Gamma waves yielded the highest accuracy, but do not provide a sufficient explanation. The authors should discuss the specific characteristics of Gamma waves and their relevance to emotion processing. Provide citations to neurophysiological studies that support the observed results.
- Inclusion of Statistical Significance in Table 2
- Table 2 lacks information on statistical significance, which is crucial for interpreting the results. The authors should include significance values in the table, which leads to the conclusions in Line 383.
- Detailed Explanation of Results (Lines 398-399):
- Similar to line 333, a clear explanation is needed. Expand on the interpretation of the results, providing a more detailed analysis of the findings and their implications.
Suggestions for Future Work:
- Combining feature extraction using Stockwell entropy with the Common Spatial Pattern (CSP) with Clustering and then Classification could potentially yield better results.
Author Response
Thank you very much for taking the time to review this manuscript.
Comments 1: [Preprocessing Data
Inconsistent Bandpass Filtering (Lines 116-117)
Line 116-117: The paper states that the data was "filtered with a 0-75 Hz bandpass filter," which differs from reference document #13, which used "0.3-50 Hz."
The authors should provide a detailed justification for this difference. If the deviation is intentional, the rationale and potential impact on the data should be thoroughly explained. If it was an error it must be corrected. Suboptimal Power Line Noise Removal (Line 118):
Line 118: The authors claim to have used a "bandpass filter 0-46 Hz" to avoid the 50 Hz power line noise. This reasoning is flawed. The original data could have been used with the addition of a "notch filter 50 Hz."]
Response 1: [Thank you for pointing this out. Both the paper and citations utilize the SEED public database from Shanghai Jiao Tong University. The EEG data in the database has been downsampled to 200 Hz and filtered with a 0–75 Hz bandpass filter. In the preprocessing section, the cited paper applied an additional bandpass filter to the raw data, with a frequency range of 0.3 Hz to 50 Hz. According to the usage agreement for the SEED database, this article must be cited.
In our paper, the frequency range was set between 0.1 Hz and 46 Hz, but it was mistakenly written as 0–46 Hz. We appreciate your reminder, and we have corrected this error. Additionally, we have removed the phrase "to avoid interference from the 50 Hz power line noise."]
Comments 2:[ Common Spatial Patterns (CSP)
Lack of Component Selection Justification (Lines 187-188):
The authors select 8 CSP components from 62, but fail to provide a clear explanation of the selection process. This raises questions about the optimality of the chosen components and their impact on the results. In other words, the authors do not explain how the 8 CSP components were selected from the 62 components, nor whether this selection is sufficient. This creates a disconnect with Figure 8. Additionally, reference #41 uses ICA, not CSP.
Missing Link to Figure 8:
The link between the number of CSP components selected and the information displayed in figure 8 is not clear.]
Response 2:[ Thank you for your reminder regarding the citation error. We have now corrected it to the proper reference number.
The core objective of the CSP is to extract features that distinguish between different classes of tasks from multi-channel EEG data. This is achieved by maximizing the variance differences between the two classes of signals, thereby obtaining highly discriminative feature vectors that also serve a dimensionality reduction purpose.
Thus, the 8 CSP components represent the optimal spatial filters derived from the computation of the 62-channel EEG signals. In our experiments, we tested various numbers of components and found that selecting 8 achieves a relatively satisfactory performance.]
Comments 3:[ Feature Extraction Process:
Unexplained Epoch Length Choice (Line 195):
The authors do not explain the rationale for choosing an epoch length of 3 seconds, especially since the source article (#13) used an epoch length of 1 second. This requires justification.]
Response 3:[ A good question. Unlike event-related EEG signals, emotional states require a period of EEG data to be effectively expressed. Therefore, in the experiments conducted by Shanghai Jiao Tong University, participants were shown movie clips lasting 3 to 4 minutes to elicit different emotional states. Currently, there is no specific research determining the optimal duration for each epoch (e.g., 1 second, 2 seconds, or 3 seconds). Theoretically, longer durations may be more advantageous as they allow for more complete emotional expression. In our experiments, using epochs of 3 seconds achieved a good balance between computational efficiency and performance.
Thank you for your suggestion. We can consider conducting dedicated research on this aspect in the future.]
Comments 4:[ Classification:
Insufficient Explanation of Pairwise Classification (Lines 219-221) : The rationale for using pairwise classification across three emotion groups is not clearly explained. Readers may not understand why pairwise verification is necessary across three groups when a ternary classification of all three emotions could be used.]
Response 4:[ 1. The SEED dataset provides three emotional categories: positive, negative, and neutral. The choice between two-class and three-class classification can be made depending on the specific research objectives. Some studies opt for binary classification, while others choose three-class classification.
- Although the CSP (Common Spatial Pattern) algorithm is more suitable for binary classification tasks, we applied it to a three-class classification scenario and found that it still achieved satisfactory results.]
Comments 5:[ Results Presentation:
Clarity in Table 1 (Line 248): Line 248, Table 1: Use abbreviations (e.g., Pos-Neg, Pos-Neu) instead of numerical codes for clarity. Also, brainwaves should be listed in order of frequency.
Figure 4-6: Replace the bar graphs in Figures 4-6 with confusion matrices to provide a more detailed and easily interpretable representation of the classification performance.]
Response 5:[ Thank you for your suggestion. We have made corrections accordingly.
As you pointed out, using a confusion matrix is indeed a better approach. We also considered utilizing a confusion matrix; however, since our study involves comparing the recognition rates of different classification algorithms across various frequency bands, adopting confusion matrices would require a large number of figures. Therefore, we found that bar charts provide a more comprehensive and concise representation for this purpose.]
Comments 6:[ Explanation of Gamma Wave Accuracy (Line 333)
The authors report that Gamma waves yielded the highest accuracy, but do not provide a sufficient explanation. The authors should discuss the specific characteristics of Gamma waves and their relevance to emotion processing. Provide citations to neurophysiological studies that support the observed results.
Inclusion of Statistical Significance in Table 2
Table 2 lacks information on statistical significance, which is crucial for interpreting the results. The authors should include significance values in the table, which leads to the conclusions in Line 383.
Detailed Explanation of Results (Lines 398-399):
Similar to line 333, a clear explanation is needed. Expand on the interpretation of the results, providing a more detailed analysis of the findings and their implications.]
Response 6:[ The question you raised is highly valuable. It has been previously reported in earlier studies that gamma waves achieve the highest accuracy. However, the focus of this paper is on methodological improvements that provide better recognition rates. As for why gamma brain waves are closely related to emotions, there is currently no definitive conclusion in existing research. Some papers exploring emotion-related EEG methods have also not delved further into the physiological reasons behind this.
The question you raised is indeed significant, and we will address it as part of our future work.
Regarding Table 2, its primary purpose is to compare the standard deviations of the Stockwell entropy of the signal sin(2πft) under different frequencies and window widths. It demonstrates that as frequency increases, the standard deviation generally decreases, and the values remain relatively stable, leading to better classification performance. This analysis is not intended for hypothesis testing; therefore, we believe that adding significance tests is unnecessary in this context.]
Comments 7:[ Suggestions for Future Work:
Combining feature extraction using Stockwell entropy with the Common Spatial Pattern (CSP) with Clustering and then Classification could potentially yield better results.]
Response 7:[ An excellent suggestion, thank you for your guidance. We will subsequently explore the application of combining Stockwell entropy with clustering.]
Thank you for providing many valuable opinions.
Reviewer 2 Report
Comments and Suggestions for Authors
The research problem formulation is not precise in the introduction. Highlight the contributions of this paper.
Through analysis of Stockwell entropy for the different parameters to be analysed, including length.
State the core works used in this article with proper citation. Show how Stockwell entropy differs from the original work.
Stockwell Entropy - Literature is available - line 150.
https://doi.org/10.1016/j.patcog.2020.107687
Check the math notations, for example, line 174.
Rewrite the equations with the proper notations for matrix (bold(X)), vector (bold(x)), and scalar (x) parameters. Avoid ambiguity in expressions, for example, N in line 163, N - length of signal.
Lines 227-229 mention two classifiers, but the reasons for selecting these methods were not presented.
Comments on the Quality of English LanguageAbstract: Mention two-class tasks such as "positive Vs neutral", "negative Vs neutral", and "positive Vs negative."
Avoid statements that don't add value to the abstract. For example, "This study provides a new idea for EEG-based emotion recognition."
Introduction: Check the symbols used in line 31
Don't define the same DE for two entropy measures such as described in lines 45, 46
Check line 48: spectral energy (ES)?? Check line 435.
Check line 61 - common space pattern (CSP) ?
Related work:
Don't define the terms again and again... CSP! @ line 77, line 95, 177, 331. 137&76, even in conclusion!
Avoid repeated lines, for example, lines 68-70, 94-95, the first line of the abstract, and line 327!!
Methods:
Poor formatting... why line 161 is bold? Annotate the figures properly, for example fig.2 - csp components of what? Also, what does a different colour level indicate?
First, define the words as they appear first. For example, SVM is stated in 186, 208, and defined in 227, and redefined at 444!!
The same is already at line 156.
Parameter k is not vector @ line 164.
Be consistent in the notations W5 or W_5. Refer to Table 1 and the rest of the test in the results section.
Check Fig 6 - is it CPS or CSP?
Results and discussions: well presented, however, avoid repeated statements.
Author Response
Thank you very much for taking the time to review this manuscript.
Comments 1: [The research problem formulation is not precise in the introduction. Highlight the contributions of this paper.
Through analysis of Stockwell entropy for the different parameters to be analysed, including length.
State the core works used in this article with proper citation. Show how Stockwell entropy differs from the original work.]
Response 1: [Thank you for your suggestion. We have revised the phrasing of the research questions in the introduction on page two accordingly.]
Comments 2: [Stockwell Entropy - Literature is available - line 150.
https://doi.org/10.1016/j.patcog.2020.107687]
Response 2: [Thank you for your suggestion. We have already cited the paper. ]
Comments 3: [Check the math notations, for example, line 174.
Rewrite the equations with the proper notations for matrix (bold(X)), vector (bold(x)), and scalar (x) parameters. Avoid ambiguity in expressions, for example, N in line 163, N - length of signal.]
Response 3: [Thank you for your corrections. We have already fixed these errors.]
Comments 4: [Lines 227-229 mention two classifiers, but the reasons for selecting these methods were not presented.]
Response 4: [Thank you for your guidance. We have added the reasons for choosing these two methods on page six.]
Comments 5: [Abstract: Mention two-class tasks such as "positive Vs neutral", "negative Vs neutral", and "positive Vs negative."]
Response 5: [Thank you for your guidance. The corresponding revisions have been made in all relevant sections of the paper.]
Comments 6: [Avoid statements that don't add value to the abstract. For example, "This study provides a new idea for EEG-based emotion recognition."]
Response 6: [Thank you for your guidance. It has been deleted.]
Comments 7: [Introduction: Check the symbols used in line 31
Don't define the same DE for two entropy measures such as described in lines 45, 46
Check line 48: spectral energy (ES)?? Check line 435.
Check line 61 - common space pattern (CSP) ?
Related work:
Don't define the terms again and again... CSP! @ line 77, line 95, 177, 331. 137&76, even in conclusion!
Avoid repeated lines, for example, lines 68-70, 94-95, the first line of the abstract, and line 327!!
]
Response 7: [Thank you for your guidance. The relevant content has been modified or deleted.]
Comments 8: [Methods:
Poor formatting... why line 161 is bold? Annotate the figures properly, for example fig.2 - csp components of what? Also, what does a different colour level indicate?
First, define the words as they appear first. For example, SVM is stated in 186, 208, and defined in 227, and redefined at 444!!
The same is already at line 156.
Parameter k is not vector @ line 164.
Be consistent in the notations W5 or W_5. Refer to Table 1 and the rest of the test in the results section.
Check Fig 6 - is it CPS or CSP?]
Response 8: [Thank you for your guidance. The relevant content has been modified or deleted.]
Comments 9: [Results and discussions: well presented, however, avoid repeated statements.]
Response 9: [Thank you for your affirmation.]
Thank you for providing many valuable opinions.
Reviewer 3 Report
Comments and Suggestions for Authors
In this study, the authors combined differential entropy computation on data after Stockwell transformation (ST) and common spatial pattern (CSP) approach for distinguishing different emotional states. Although the results of the investigation can be of interest in the emotional recognition field as these highlight the involvement of high-frequency neural activity (gamma band) in emotional states, in my opinion the paper as a whole is not suitable for publication because of the lack of rigor in the structural organization, the weak technical language in the methodological part, the redundancy in the presentation of the results, and the lack of clarity in the discussion of the results. For these reasons, I suggest to reject the work.
Some main comments are given below:
- The introduction presents a basic language and reports the repetition of some concepts without giving a clear idea of the improvement given by the presented research. The general information provided at the beginning of section 2, related to the advantages of using ST and CSP, should be moved in the introduction, as these tools are only mentioned in the last part of the introduction without giving an idea of the improvement related to their usage. Moreover, in my opinion, the several reported examples related to the application of both approaches also in other fields can be used in the discussion session. Regarding this part, it is difficult to follow the text because of the given details often indicated by several not defined acronyms.
- As regards the methodological part, several concerns are about the parameters used for performing the several steps of the analysis from the application of Stockwell Transform, e.g. no details are given on the changing width of the gaussian windows, to the overlap of the windows used to obtain entropy measures or its width (only in the results section it was reported that that parameter was changed in the analysis, but it is not clear its value as it is expressed in a percentage of a not explicitly expressed time). Moreover, the parameters used in the classification analysis should be expressed in paragraph 3.4, as also to avoid a repetition of the analysis performed at the beginning of each paragraph of the results section. Another point regards the use of the eight groups of features obtained through the CSP algorithm and that allows to have a spatial distribution of the investigated features. How is the information obtained from the different channels combined in facing the classification problem? In my opinion, maintaining the information on the spatial distribution of the entropy features can be useful in detecting also the mainly involved cortical region in emotions recognition.
- All the results are redundantly reported both in table, figures and in the text, as only two figures (one for SVM and the other for RF) could be useful in depicting all the results in all bands for the four classification settings (negative-neutral, positive-neutral, negative-positive and negative-positive-neutral).
- Once arrived at the end of section 4, the reader expects a discussion of the findings, but instead several preliminary analyses are presented, which, if they are to be retained (albeit in a more restricted manner in my opinion) should be reported in the methodological part. Moreover, it is not clear for me the reason why to investigate the role of the width of the window and not of other parameters of the analysis.
- In line 150 it is stated ‘we define this as Stockwell Entropy’, but from a fast search in the literature this term is already used as the application of entropy measures on signals transformed with the S-transform. Same comment for lines 328-330.
- The discussion section should be improved by considering also the physiological meaning of the findings, also considering the relevance of other frequency bands e.g. beta. Moreover, the sentence in lines 395-397 should be rephrased, because following the reported reasoning, the high-frequency band is always the most stable and thus the most suitable to detect changes in the state of the subjects.
General comments:
- Some acronyms are constantly defined in the whole text, e.g. CSP, DE, etc…
- The acronym DE was used for both differential entropy and dispersion entropy, while the ES acronym refers to energy spectrum and not spectral energy.
- Equations are inserted as figures.
In my opinion, the english could be improved to more clearly express the research.
Author Response
Thank you for your insightful suggestions. We have made the corresponding revisions based on your advice, and we also hope to discuss some points with you further.
Comments 1: [The introduction presents a basic language and reports the repetition of some concepts without giving a clear idea of the improvement given by the presented research. The general information provided at the beginning of section 2, related to the advantages of using ST and CSP, should be moved in the introduction, as these tools are only mentioned in the last part of the introduction without giving an idea of the improvement related to their usage. Moreover, in my opinion, the several reported examples related to the application of both approaches also in other fields can be used in the discussion session. Regarding this part, it is difficult to follow the text because of the given details often indicated by several not defined acronyms.]
Response 1: [Thank you for your valuable feedback. We have removed some of the redundancies related to conceptual repetition.
Since this paper primarily focuses on combining Stockwell entropy with CSP for EEG-based emotion recognition, our work differs from previous studies by applying Stockwell entropy for feature extraction. In the discussion section, we emphasize the impact of Stockwell entropy on different EEG frequency bands. Therefore, we believe it is more appropriate to introduce the learning framework of entropy measures in the Introduction section.]
Comments 2: [As regards the methodological part, several concerns are about the parameters used for performing the several steps of the analysis from the application of Stockwell Transform, e.g. no details are given on the changing width of the gaussian windows, to the overlap of the windows used to obtain entropy measures or its width (only in the results section it was reported that that parameter was changed in the analysis, but it is not clear its value as it is expressed in a percentage of a not explicitly expressed time). Moreover, the parameters used in the classification analysis should be expressed in paragraph 3.4, as also to avoid a repetition of the analysis performed at the beginning of each paragraph of the results section. Another point regards the use of the eight groups of features obtained through the CSP algorithm and that allows to have a spatial distribution of the investigated features. How is the information obtained from the different channels combined in facing the classification problem? In my opinion, maintaining the information on the spatial distribution of the entropy features can be useful in detecting also the mainly involved cortical region in emotions recognition.]
Response 2: [Thank you for your reminder. In the experiment, the Stockwell transform of EEG was performed using the MNE analysis library, and we have now added a description of the parameters in the paper. The choice of 8 CSP components was based on experimentation with different numbers, and we found that selecting 8 achieved a relatively satisfactory performance.
Regarding your question, " How is the information obtained from the different channels combined in facing the classification problem? In my opinion, maintaining the information on the spatial distribution of the entropy features can be useful in detecting also the mainly involved cortical region in emotions recognition."
Your question is highly valuable and this is indeed an aspect that our current study did not consider. However, we will take it into account in our future research.]
Comments 3: [All the results are redundantly reported both in table, figures and in the text, as only two figures (one for SVM and the other for RF) could be useful in depicting all the results in all bands for the four classification settings (negative-neutral, positive-neutral, negative-positive and negative-positive-neutral).]
Response 3: [Thank you for your suggestion. We initially considered minimizing the number of figures to streamline the presentation. However, since the comparison also involves different EEG frequency bands, we ultimately decided to use four figures to ensure clarity and comprehensiveness in the presentation.]
Comments 4: [Once arrived at the end of section 4, the reader expects a discussion of the findings, but instead several preliminary analyses are presented, which, if they are to be retained (albeit in a more restricted manner in my opinion) should be reported in the methodological part. Moreover, it is not clear for me the reason why to investigate the role of the width of the window and not of other parameters of the analysis.]
Response 4: [According to the definition of Stockwell entropy, if the data is assumed to follow a Gaussian distribution, only the width of the sliding window needs to be considered.
We would like to discuss this point with you. Since this paper primarily focuses on exploring the classification of emotion-related EEG from the perspective of data analysis, the discussion section centers on the methodology. Through experiments, we first demonstrate that the combination of Stockwell entropy and the CSP method achieves better recognition rates. Subsequently, we delve into the impact of Stockwell entropy on the signal. This structure appears to be more appropriate for presenting our findings.]
Comments 5: [In line 150 it is stated ‘we define this as Stockwell Entropy’, but from a fast search in the literature this term is already used as the application of entropy measures on signals transformed with the S-transform. Same comment for lines 328-330.]
Response 5: [Thank you for your reminder. We were not previously aware of the relevant paper, but we have now made the necessary revisions to our manuscript and added the appropriate citation..]
Comments 5: [The discussion section should be improved by considering also the physiological meaning of the findings, also considering the relevance of other frequency bands e.g. beta. Moreover, the sentence in lines 395-397 should be rephrased, because following the reported reasoning, the high-frequency band is always the most stable and thus the most suitable to detect changes in the state of the subjects.]
Response 5: [This is an excellent suggestion. As mentioned earlier, since this paper primarily focuses on exploring the classification of emotion-related EEG from the perspective of data analysis, the discussion section concentrates on the methodology. We reviewed similar articles and found that many papers do not delve into the underlying physiological reasons either. This aspect can be considered as part of our future work.]
Comments 6: [Some acronyms are constantly defined in the whole text, e.g. CSP, DE, etc…
The acronym DE was used for both differential entropy and dispersion entropy, while the ES acronym refers to energy spectrum and not spectral energy.]
Response 6: [Thank you for your corrections. We have revised or removed the corresponding content accordingly..]
Comments 7: [Equations are inserted as figures.]
Response 7: [Thank you for your feedback. We have made the necessary corrections.
Previously, we used the MathType Office plugin, which converted all formulas into images upon saving. We have now revised all the equations in the paper using the MathType software.]
Thank you for providing many valuable opinions.